# Insight into postural control in unilateral sensorineural hearing loss and vestibular hypofunction

Anat V. Lubetzky[1]*, Jennifer L. Kelly[2], Daphna Harel[3], Agnieszka Roginska[4], Bryan D. Hujsak[2], Zhu Wang[5], Ken Perlin[5], Maura Cosetti[6]

1 Department of Physical Therapy, Steinhardt School of Culture, Education and Human Development, New York University, New York, NY, United States of America, 2 Vestibular Rehabilitation, New York Eye and Ear Infirmary of Mount Sinai, New York, NY, United States of America, 3 Department of Applied Statistics, Social Science and Humanities, Steinhardt School of Culture Education and Human Development, New York University, New York, NY, United States of America, 4 Department of Music and Performing Arts Professions, Steinhardt School of Culture, Education and Human Development, New York University, New York, NY, United States of America, 5 Computer Science Department, Courant Institute of Mathematical Sciences, New York University, New York, NY, United States of America, 6 Department of Otolaryngology-Head and Neck Surgery, New York Eye and Ear Infirmary of Mount Sinai, New York, NY, United States of America

* anat@nyu.edu

**Data Availability Statement:** The data are available here: https://data.mendeley.com/datasets/kcvbs468rm/1 To cite this dataset: Lubetzky, Anat

## Abstract

This pilot study aimed to identify postural strategies in response to sensory perturbations (visual, auditory, somatosensory) in adults with and without sensory loss. We tested people with unilateral peripheral vestibular hypofunction (N = 12, mean age 62 range 23–78), or with Unilateral Sensorineural Hearing Loss (USNHL, N = 9, 48, 22–82), or healthy controls (N = 21, 52, 28–80). Postural sway and head kinematics parameters (Directional Path in the anterior-posterior and medio-lateral directions (sway & head); pitch, yaw and roll (head) were analyzed in response to 2 levels of auditory (none, rhythmic sounds via headphones), visual (static, dynamic) and somatosensory cues (floor, foam) within a simulated, virtual 3-wall display of stars. We found no differences with the rhythmic auditory cues. The effect of foam was magnified in the vestibular group compared with controls for anterior-posterior and medio-lateral postural sway, and all head direction except for medio-lateral. The vestibular group had significantly larger anterior-posterior and medio-lateral postural sway and head movement on the static scene compared with controls. Differences in pitch, yaw and roll emerged between vestibular and controls only with sensory perturbations. The USNHL group did not increase their postural sway and head movement with the increased visual load as much as controls did, particularly when standing on the foam. They did not increase their medio-lateral sway with the foam as much as controls did. These findings suggest that individuals with USNHL employ a compensatory strategy of conscious control of balance, the functional implications of which need to be tested in future research.

(2022), "Hearing Health Foundation ERG project", Mendeley Data, V1, doi: 10.17632/kcvbs468rm.1.

**Funding:** This study was funded by the Hearing Health Foundation Emerging Research Grant 2019. https://hearinghealthfoundation.org/how-to-apply AL, DH, BH and MC are also funded by a grant from the National Institute on Deafness and other Communication Disorders. The sponsors had no role in the study design, collection, analysis and interpretation of data; in the writing of the manuscript; or in the decision to submit the manuscript for publication.

**Competing interests:** The authors have declared that no competing interests exist.

**Abbreviations:** USNHL, Unilateral Sensorineural Hearing Loss; VH, Vestibular Hypofunction; VR, Virtual Reality; DHI, Dizziness Handicap Inventory; ABC, Activities Specific Balance Confidence Scale; SSQ12, The Speech, Spatial and Quality of Hearing 12-item Scale; SSQ, Simulator Sickness Questionnaire.

## Introduction

Recent studies have demonstrated a relationship between hearing loss and increased risk of falls, slower walking speed, balance instability and significantly poorer performance on standardized fall-risk assessment tools [1–11]. However, the implications for different types of hearing loss and the mechanism underlying the relationship between hearing loss and balance dysfunction is still debatable.

Prior studies in adults focused on individuals with bilateral hearing loss or a mix of unilateral and bilateral loss [1]. Very few studies focused on individuals with severe, unaidable, unilateral sensorineural hearing loss (USNHL) with normal or near normal hearing in the other ear [12]. Wolter et al [13]. found poorer balance performance in children with single-sided hearing loss compared to children with normal hearing, particularly on challenging tasks, such as single leg stance with eyes closed. Individuals with USNHL are unique because they hear normally in a quiet environment with limited sound sources but cannot localize the sound. While historically felt to present no functional limitations, data now suggests that USNHL may lead to participation restrictions in social, family and work settings [14, 15]. USNHL has reported prevalence between 20 to 160 cases for every 100,000 people and an annual incidence of over 4,000 in the US alone [16, 17]. Yet the effect of severe USNHL on balance is not well understood and is a missing piece in current clinical practice.

When healthy individuals perform balance tasks in complex sensory environments, they tend to respond to sensory perturbations (e.g., moving visual environments) by increasing their body sway [18–20]. The "auditory anchor" theory suggests that individuals use auditory spatial cues for balance, analogous to visual cues [21]. According to this theory, information from stationary sound sources creates a spatial map of the environment which is used for stabilization; this information may be missing or unreliable in individuals with hearing loss. Alternatively, several authors suggested that balance dysfunction in patients with hearing loss may arise from concomitant vestibular dysfunction even without vestibular symptoms [22, 23]. Both aforementioned theories suggest individuals with hearing loss may demonstrate increased instability in busy visual environments or on unstable surfaces similar to adults with vestibular loss. Another possible mechanism suggests that individuals with hearing loss compensate for the loss of auditory cues by utilizing a "feed-forward" mechanism for postural control, i.e., relying on prior expectation and motor planning rather than responding to dynamic sensory cues. [24]. Lack of normal increase in body sway with sensory perturbations, i.e., conscious movement processing strategy, has been exemplified in people with Persistent Postural-Perceptual Dizziness [25] but has not been tested directly in adults with USNHL. This pilot study aimed to identify postural strategies in response to sensory perturbations (visual, auditory, somatosensory) in with individuals with USNHL (and no vestibular complaints), individuals with vestibular hypofunction, and age-matched controls. We used a well-established protocol [26, 27] where we added an auditory layer.

## Materials and methods

### Sample

Adult participants (18 or older) with unilateral peripheral vestibular hypofunction and no hearing loss, i.e., vestibular neuritis, or with sudden idiopathic unilateral sensorineural hearing loss (USNHL), no evidence of retrocochlear pathology on MRI and no vestibular complaints, were recruited prospectively from the New York Eye and Ear Infirmary of Mount Sinai. Unilateral vestibular hypofunction was determined based on positive findings on bedside clinical exam (positive head thrust test, post head shaking nystagmus or spontaneous nystagmus) and / or

**Table 1. Virtual reality postural control protocol.**

|  | Visual | Auditory | Somatosensory |
|---|---|---|---|
| Low sensory load | Static | No sound | Floor |
| High sensory load | Stars moving AP at 0.2 Hz, 0.032m [26] | white noise cycles, i.e., intensity increases from 0 to 3dB at 0.3Hz | Airex Balance Pad 20"L x 16.4"W x2.5"H. |

horizontal canal paresis > 25% if caloric testing was available [28]. USNHL was defined as normal hearing (<26 dB 4 frequency (0.5, 1, 2 and 4 kHz) pure tone average) in one ear and at least moderate (> 41 dB 4 frequency PTA) SNHL in the affected ear [29]. Control participants were recruited from the community. All participants were screened for normal or corrected to normal vision, normal protective sensation at the bottom of their feet and reported no neurological conditions. No participants used a cane or walker. Control participants self-reported normal hearing and no vestibular symptoms. No participants in this study wore any kind of amplification, including traditional hearing aids, CROS (contralateral routing of signals) or Bi-CROS aids. Participants in the USNHL were excluded for mixed or conductive hearing loss.

## Testing protocol

Postural sway and head kinematics parameters were analyzed in response to 2 levels of auditory, visual and somatosensory cues within a virtual stars scene as detailed in Table 1. The visual protocol is well-established [26, 27] and we added an auditory layer of a rhythmic sound to match the oscillating visual perturbations (see S1 Video) in a district frequency, both comfortable frequencies for human standing postural control (0.2 or 0.3 Hz) [30, 31].

This study was approved by the Institutional Review Board of the Icahn School of Medicine at Mount Sinai (#STUDY-18-00431) and New York University Committee on Activities Involving Research Subjects (#IRB-FY2016-155). The individual in this manuscript has given written informed consent (as outlined in PLOS consent form) to publish these case details (Fig 1). All participants first signed a written informed consent. Then their vision was screened using the standard EDTRS chart in ambient room light. The acceptable standard for driving as our measure for adequate vision was applied at a visual acuity of 20/60. Participants could wear glasses for testing, so long as they were worn for the postural assessment as well. Normal protective sensation at the bottom of the feet was confirmed by the ability to perceive a 5.07 (10 gram) monofilament [32]. The participant's height and forehead length were measured to ensure that the visual scene would be projected at their eye height. Participants completed the Simulator Sickness Questionnaire [33] at baseline, during breaks and at the end of the session. This 16-item scale asks participants to rate symptoms such as dizziness, blurred vision or fatigue on a scale of 0 (none) to 3 (severe). For the virtual reality (VR) protocol, participants stood hip-width apart on a stable force-platform (Kistler 5233A, Winterthur, Switzerland) wearing the Vive Pro Eye headset (HTC Corporation, Taoyuan City, Taiwan). Participants were asked to look straight ahead and do whatever felt natural to them to maintain their balance. There were 4 possible combinations of visuals / sounds: static visual / no sound; static visual / rhythmic white noise; moving visuals / no sound; moving visuals/ rhythmic white noise. Each trial lasted 60 seconds and was repeated 3 times on each surface for a total of 24 conditions: 12 standing directly on the stable force-platform embedded in the floor and 12 standing on Airex memory foam placed on top of the force-platform. The order of the conditions was randomized per block meaning that the same randomized order was performed on floor or foam (Table 1 and S1 Video). For safety reasons, the floor block was always presented first. Participants were guarded by either a licensed or student physical therapist. Two quad

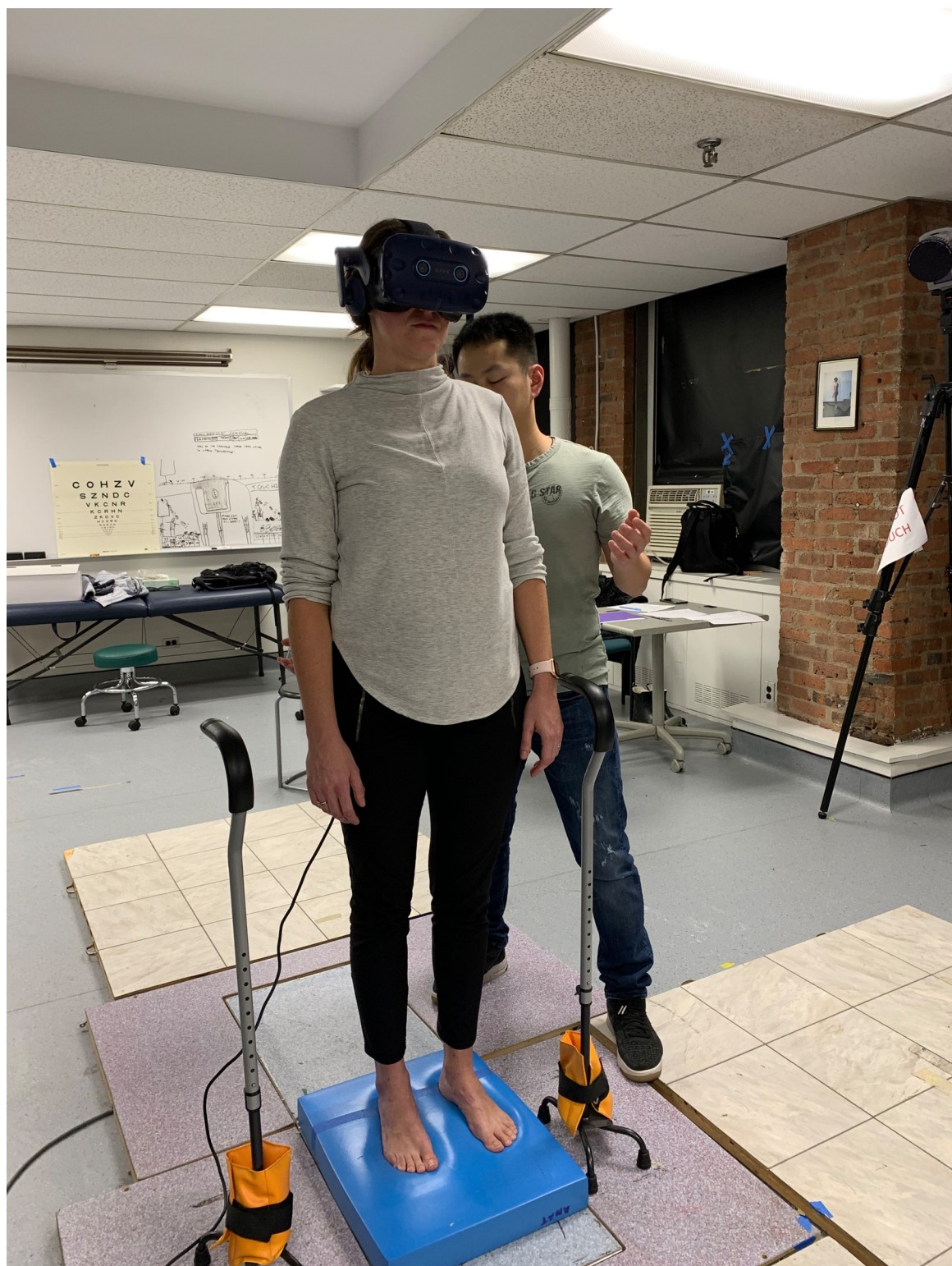

**Fig 1. Experimental setup when a participant is standing on memory foam.**

canes were placed on either side of the force-platform for safety and to help with stepping on and off the force-platform, particularly with the foam (See Fig 1). A rest break was provided after 12 trials but offered as needed. During the break, participants completed several questionnaires: The Dizziness Handicap Inventory (DHI) [34, 35]; The Activities-Specific Balance Confidence (ABC) [35]; and the Speech, Spatial and Quality of Hearing 12-item Scale [SSQ12] [36]. The SSQ12 was only administered to those in the USNHL group (see Table 2).

**Table 2. Description of the sample.**

| | Control Group | Vestibular Hypofunction | USNHL | P value |
|---|---|---|---|---|
| Age (years): mean (min, max, SD) | 52 (28, 80, 16.92) | 62 (23, 78, 17.95) | 48 (22, 82, 19) | P = 0.16$^\diamond$ |
| Gender (N and % females) | 13 (62%) | 10 (83.3%) | 4 (44.4%) | P = 0.18$^{\diamond\diamond}$ |
| Falls in the past year (count) | None = 18 | None = 10 | None = 8 | P = 0.57$^{\diamond\diamond}$ |
| | One = 3 | One = 1 | One = 1 | |
| | | Ten = 1 | | |
| Race | White = 16 (76.2%) | White = 7 (58.3%) African American = 4 (33.3%) | White = 2 (22.2%) | P = 0.03$^{\diamond\diamond}$ |
| | African American = 3 (14.3%) | Latino = 1 (8.3%) | African American = 2 (22.2%) | |
| | Latino = 1 (4.8%) | | Latino = 2 (22.2%) | |
| | Other = 1 (4.8%) | | Asian American = 3 (33.3%) | |
| Weight (Kg): mean (min, max, SD) | 74.88 (54.4, 99.8, 12.68) | 79.03 (60.32, 98.9, 10.72) | 75.19 (57.15, 120.0, 19.16) | P = 0.36$^{\diamond\diamond\diamond}$ |
| Height (cm): mean (min, max, SD) | 169.56 (157.48, 180.34, 6.24) | 165.63 (154.94, 182.88, 7.87) | 169.17 (156.21, 182.88, 8.07) | P = 0.3$^\diamond$ |
| Exercise: yes / no count | Yes = 20 | Yes = 9 | Yes = 6 | P = 0.12$^{\diamond\diamond}$ |
| | No = 1 | No = 2 | No = 3 | |
| | | Missing = 1 | | |
| Exercise (minutes per week): mean (min, max, SD) | 405.43 (0, 1275, 343.32) | 175.00 (0, 630, 175.73) | 193.00 (0, 560, 206.12) | P = 0.038$^{\diamond\diamond\diamond}$ Vestibular < controls adjusted sig. 0.056 |
| Dizziness Handicap Inventory (DHI): mean (min, max, SD) | 0 (0, 0, 0) | 43.33 (28, 74, 12.6) | 5.33$^\#$ (0, 32, 11.31) | P<0.001$^{\diamond\diamond\diamond}$ Vestibular > than controls and > than USNHL adjusted significance P<0.001 |
| Activities-Specific Balance Confidence (ABC): mean (min, max, SD) | 97.48 (77.18, 100, 5.39) | 66.23 (32.5, 95.93, 23.96) | 96.19 (79.3, 100, 6.64) | P<0.001$^{\diamond\diamond\diamond}$ Vestibular > than controls and > than USNHL adjusted significance P<0.001 and P = 0.003 respectively |
| The Speech, Spatial and Quality of Hearing 12-item Scale (SSQ12): mean (min, max, SD) | NA | NA | 5.81 (4, 7, 0.86) | |
| Simulator Sickness Questionnaire Pre: mean (min, max, SD) | 0.38 (0, 3, 0.81) | 3.33 (0, 11, 2.71) | 1 (0, 6, 2) | P<0.001$^{\diamond\diamond\diamond}$ Vestibular > than controls and > than USNHL adjusted significance P<0.001 and P = 0.003 respectively |
| Simulator Sickness Questionnaire Post: mean (min, max, SD) | 0.95 (0, 6, 1.56) | 5.17 (0, 30, 8.27) | 0.78 (0, 6, 1.99) | P = 0.007$^{\diamond\diamond\diamond}$ Vestibular > than controls and > than USNHL adjusted significance P = 0.008 and P = 0.004 respectively |

$^\diamond$One way ANOVA

$^{\diamond\diamond}$Chi-square for proportions

$^{\diamond\diamond\diamond}$Kruskal-Wallis

$^\#$One participant in the USNHL group had DHI = 16, another DHI = 32 (both considered mild/ borderline mild) [34] and the remaining 7 participants reported 0.

The DHI was designed to identify difficulties that a patient may be experiencing because of their dizziness. The ABC scale is a subjective measure of confidence in performing various ambulatory activities without falling or feeling 'unsteady'. The SSQ12 is valid and provides insight into the day-to-day impact of hearing loss.

## System

Visuals were implemented in C# language using standard Unity Engine version 2019.4.16f1 (64-bit) (©Unity Technologies, San Francisco, CA, USA). The scenes (visuals and sounds) were delivered via an HTC Vive Pro Eye headset (HTC Corporation, Taoyuan City, Taiwan) controlled by a Dell Alienware laptop 15 R3 (Round Rock, TX, USA). The participants observed a display of randomly distributed white spheres (diameter 0.02 meters [m]) on a black background. The display is viewed as 1.63 m away from the participant and appears to have a room arrangement with one front and two side walls, each 6.16 m by 3.2 m. A clear central area of occlusion of 0.46 m in diameter was created to suppress the visibility of "sampling artifacts" or aliasing effects, which are most likely to occur in the foveal region [37]. Height was calibrated using Steam VR standing paradigm setup.

To generate the auditory cues, we used Matlab to develop white noise in an envelope generator. We then modulated the sonic amplitude cycle from quieter to louder at 0.3 Hz. The intensity was defined as the difference between the lowest and loudest part of the oscillation. The sounds mimicked a wave where the mid-point of the cycle has the loudest point. In order to render the white noise in the Unity scenes to join the visuals, the generated audio files were further processed with Wwise which was integrated into Unity. Sounds were presented binaurally between 65–71 dB at the highest level that was comfortable to the participant. The sound level in the testing environment (for the 'no sound' scene) was measured to be between 42 dB (no air-conditioning on) to 55 dB (both ACs on). Nine participants chose 90% (67 to 70 dB): 1 USNHL, 4 controls, 4 vestibular. Five participants chose 100% (69 to 71 dB): 3 controls, 2 vestibular. All other participants listened to the sounds at 80% intensity (65 to 68 dB).

## Data reduction and outcome measures

Postural sway was recorded at 100 Hz by Qualisys software. Head kinematics at 90 Hz was recorded by custom-made software written for the HTC Vive headset [38]. The last 55 seconds (of the total 60 seconds) were used for analysis. Data processed and analyzed in Matlab R2021a (Mathworks, Natick, MA). We applied a low-pass 4th order Butterworth filter with a conservative cutoff frequency at 10 Hz [30]. **Directional Path** (DP) is a valid and reliable measure of postural steadiness and is used as an indication of how much static balance is perturbed with a sensory manipulation [39]. DP [40] was calculated as the total path length of the position curve in the medio-lateral and anterior-posterior directions for center-of-pressure data (cm) and in medio-lateral, anterior-posterior (cm) and pitch, yaw, and roll (radians) for the head data. Test-retest reliability of postural sway DP within a similar protocol without the sounds has been previously demonstrated [41]. Criterion validity of the HTC Vive to detect head kinematics in postural tasks in a comparable way to a Qualisys motion capture system has been demonstrated as well [38]. Head kinematics has been shown to be more sensitive to vestibular hypofunction than postural sway metrics [42].

## Data analysis

We ran descriptive statistics, plots and inspection of model assumptions for all outcome and descriptive measures. We conducted a visual inspection of the data to determine whether the distributions differed by sensory perturbations and observed no difference between sound conditions. To confirm this observation, we fit two linear mixed effects models for each outcome. The first included a 4-way interaction of sound by surface, visual and group. The second included only the 3-way interaction surface by visual by group. We then compared the AIC values between these models for each outcome. In all cases, AIC was lower for the model that did not include sound (See S1 Table). Therefore, sound was not included as a variable in any

of the other models. In addition, an initial inspection of the residuals from the linear mixed effects models indicated heteroskedasticity. Therefore, we log-transformed all DP data in the final models to ameliorate this issue. Following transformation all model assumptions were met.

For each of the measures of interest (DP center-of-pressure: anterior-posterior, medio-lateral. DP Head: anterior-posterior, medio-lateral, pitch, yaw, roll) we fit a linear mixed effects model. Each model included the main effects of group, visual condition, and surface condition, and their two- and three-way interactions. Group comparisons were conducted for the vestibular group vs. controls and USNHL group vs. controls. This model maximizes the information we can obtain from the data by accounting for the inherent multi-level study design (person, conditions, repetitions). That is, since each person completes various trials for each condition, the linear mixed effects model accounts for these sources of variability [43]. P-values for the fixed effects were calculated through the Satterthwaite approximation for the degrees of freedom for the T-distribution [44].

We report the P value derived from the logarithmic model and difference in means (DIM) as well as 95% confidence intervals per condition, per group in the units of measurement through the calculation of the estimated marginal means. The DIM provides an indication of the magnitude of the difference both between and within groups and conditions. We investigate 3 types of DIMs: within-group (e.g., for controls the DIMs indicate differences within the group across conditions); between-group (e.g., for the vestibular or USNHL group the comparison to controls on the static visual/floor condition); and DIMs that represent the interaction terms–meaning how much the differences due to conditions (visual, surface) differ between groups. Figures and analysis were done in R version 3.6.1 (2019-07-05, The R project for Statistical Computing) and SPSS version 27 (IBM, SPSS Inc.)

## Results

### Sample

For a detailed description of the sample and results of the Simulator Sickness Questionnaire see Tables 2 and 3. The sample included 21 healthy controls, 12 participants with unilateral vestibular hypofunction and 9 individuals with USNHL. All participants in the control group reported no hearing impairment and no vestibular symptoms (all scored 0 on the DHI, average ABC 97%., Table 2) but it was not formally tested. All participants in the USNHL group reported no vestibular complaints (average 5 on the DHI vs. 43 for the vestibular group and 96% ABC vs. 66% for the vestibular group, Table 2) but was not submitted to vestibular tests. Two patients with vestibular hypofunction were not able to stand on foam without support hence only completed the floor segment of the protocol. All other participants completed all trials.

### Center-of-pressure (COP) (see Figs 2 and 3 and S2 Table)

With the addition on the foam to a static visual scene, controls significantly increased their medio-lateral sway (DIM = 38.8 cm, P<0.001); and anterior-posterior sway (DIM = 45.4 cm, P<0.001). The effect of increased visual load was only significant on the foam (P<0.001) but not on floor (P = 0.06) for medio-lateral sway (DIM for visuals = 10cm on foam and 1.3 cm on floor). For anterior-posterior sway, the effect of increased visual load was significant on the floor (DIM = 7.5 cm, P< 0.001) and magnified on the foam (DIM for visuals on foam = 43.8cm, P<0.001 for surface by visual interaction).

The vestibular group had significantly greater postural sway than the controls for the static visual scene on the floor: medio-lateral (between-group DIM = 9.7cm, P = 0.005);

**Table 3. Description of clinical sample (data are count).** NT = Not Tested. NA = Not Applicable.

| Criterion | Vestibular Group | USNHL group |
|---|---|---|
| Side Affected | Right = 9 | Right = 5 |
| | Left = 2 | Left = 4 |
| | Unknown = 1 | |
| Time since onset | 3–4 months = 2 | 1–2 years = 4 |
| | 6–7 months = 3 | 4–6 years = 4 |
| | 10 months = 1 | Since childhood = 1 |
| | 1 year = 2 | |
| | 2–4 years = 2 | |
| | 10–11 years = 2 | |
| Videonystagmography (VNG) | Positive = 8 | NT |
| | Negative = 2 | |
| | NT = 2 | |
| Bed-side Testing | Positive Head Thrust: 3 | NT |
| | Positive Head Shaking: 6 | |
| | Gaze-evoked Nystagmus: 3 | |
| | Spontaneous Nystagmus: 0 | |
| Vertigo at onset of hearing loss | NA | N = 3 |
| Group average of 4-freqeuncy 0.5- 4kHz Pure Tone Average (PTA)* | Right: 15 dB | Affected Ear: 75 dB |
| | Left: 17 dB | Normal Hearing Ear: 13 dB |
| Average word recognition score | Right: 97% | Affected Ear: 23% |
| | Left: 96% | Normal Hearing Ear: 98% |

*Audiograms were available for 10 out of 12 participants in the vestibular group and all participants in the USNHL group.

anterior-posterior (between-group DIM = 13.5cm, P = 0.03). The effect of foam, but not of visual load, was magnified in the vestibular group compared to controls. The vestibular group increased on average by 60.8 cm with the foam vs. floor which is 22 cm more than the control group increased (medio-lateral, P = 0.03 for group by surface interaction); and by 92 cm which is on average 46.6 cm more than controls did (anterior-posterior, P<0.001).

The USNHL group did not differ from the controls for the static scene on the floor. For medio-lateral sway, the effect of foam was different between the USNHL and controls (interaction P = 0.003). With the addition of foam, the USNHL group increased on average by 34 cm which is 4.8 cm less than controls did. For anterior-posterior sway, the effect of visual load was different between the USNHL and controls (P = 0.005 for interaction USNHL by visual on foam). With dynamic visuals, the USNHL group increased on average by 26.8 cm on foam which is 17 cm less than the change in controls. See Fig 3 for individual changes with the visual load when participants are standing on foam.

## Head (see Figs 2 and 4 and S2 Table)

With the addition of foam to a static visual scene, the controls significantly increased their head movement in all directions: medio-lateral (DIM = 22.4 cm, P<0.001); anterior-posterior (DIM = 20.4 cm, P<0.001); pitch (DIM = 0.26 rad, P<0.001); yaw (DIM = 0.29 rad, P<0.001) and roll (DIM = 0.18 rad, P<0.001). A significant change with the increased visual load on the floor was observed in anterior-posterior head movement (DIM = 5.6 cm, P<0.001) and pitch (DIM = 0.03 rad, P = 0.03). This increase was magnified on the foam (anterior-posterior DIM

**Fig 2.** Postural sway (center-of-pressure data, top) and head kinematics (head, bottom) in the anterior-posterior direction (cm, left hand side) or medio-lateral direction (cm, right hand side) for the 3 groups (Unilateral Sensorineural Hearing Loss [USNHL], vestibular hypofunction [Vestibular] and healthy, age-matched controls [Control]) across surface condition (floor or foam) and visual condition (static or dynamic).

for visuals on foam = 21.1cm, 15.5 cm greater change than on floor, interaction visual by surface P<0.001; pitch DIM = 0.14 rad, 0.11 rad greater change than on floor, interaction visual by surface P = 0.003 respectively). The effect of increased visual load was significant only when

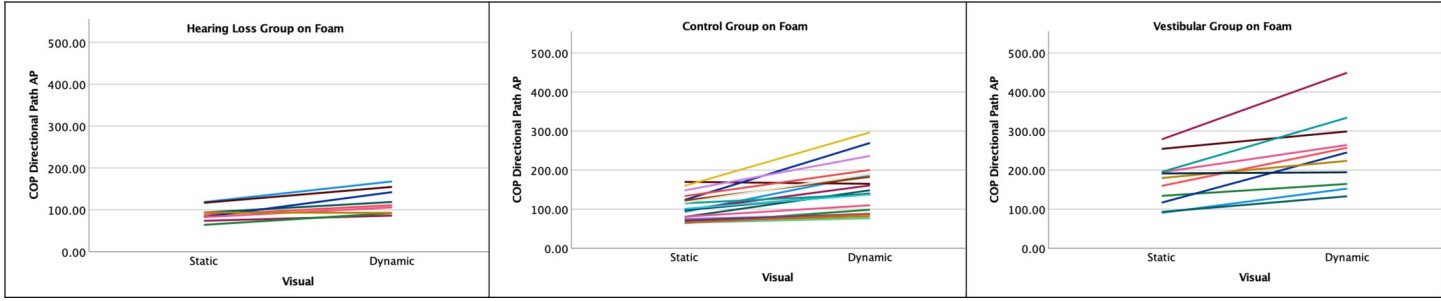

**Fig 3. Individual changes in center-of-pressure (COP) Directional Path (cm) when standing on foam and the visual environment is static or dynamic.** Each line represents one participant. The hearing loss group is one the left hand side, control group in the middle and the vestibular group on the right hand side.

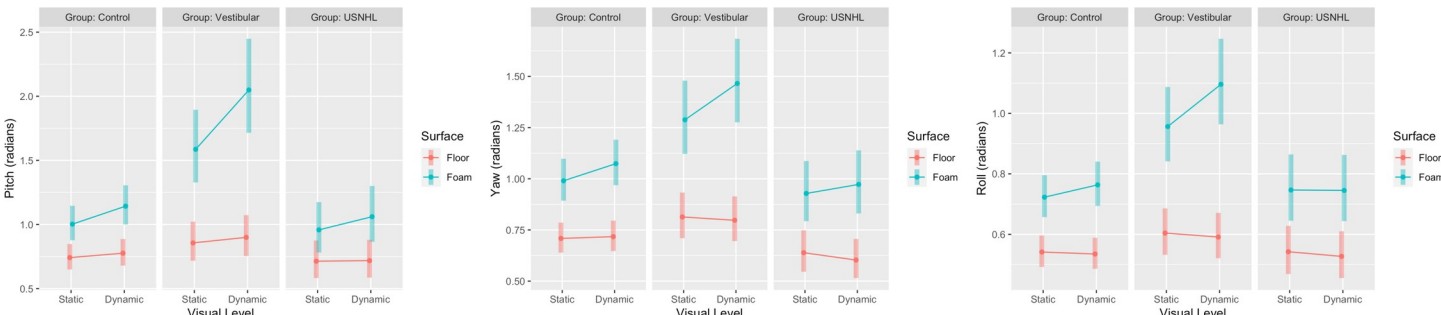

**Fig 4. Pitch, yaw and roll head movement (radians) for the 3 groups (Unilateral Sensorineural Hearing Loss [USNHL], vestibular hypofunction [Vestibular] and healthy, age-matched controls [Control]) across surface condition (floor or foam) and visual condition (static or dynamic).**

the controls were standing on the foam for the following head movement directions: medio-lateral (DIM for visuals on foam = 4.1cm, P<0.001); yaw (DIM = 0.08 rad, P<0.001) and roll (DIM = 0.04 rad, P = 0.005).

The vestibular group had significantly larger head movements than the controls for the static visual scene on medio-lateral (between-group DIM = 8.5cm, P<0.001) and anterior-posterior (between-group DIM = 10.5cm, P<0.001) but not on pitch, yaw or roll. The effect of foam was magnified in the vestibular group compared to the controls in all except for the medio-lateral direction: anterior-posterior (DIM = 36.5 cm, on average 16.1 cm greater than controls P<0.001); pitch (DIM = 0.73 rad, on average 0.47 rad greater than controls P<0.001); yaw (DIM = 0.47 rad, on average 0.18 greater than controls, P = 0.001) and roll (DIM = 0.35 rad, on average 0.17 rad greater than controls, P<0.001). The effect of increased visual load was significantly greater for the vestibular group compared to the controls on the foam only for pitch movement: DIM for visuals on foam = 0.46 rad, on average 0.32 rad greater change than controls, interaction P = 0.001).

Head movement of the USNHL group did not differ from controls on the static scene on the floor. Neither did USNHL differ from the controls in pitch, yaw or roll on any condition. For medio-lateral and anterior-posterior head movements, the effect of visual load was different between the USNHL and controls when standing on foam. The USNHL group increased their medio-lateral head movements on average by 0.2 cm which is 22.2 cm less than the change with visuals on foam in controls (interaction P = 0.04). The USNHL group increased their anterior-posterior head movements on average by 16.3 cm which is 4.8 cm less than the change with visuals on foam in controls (interaction P = 0.03).

## Discussion

Prior studies of sensory integration have demonstrated that when healthy individuals perform balance tasks in complex sensory environments, they respond to sensory perturbations (e.g., moving visual environments) with increased postural sway [18–20]. Postural control of participants in the present study mimicked published results and provided context for postural responses in those with unilateral hearing or vestibular loss. Given that balance problems related to vestibular hypofunction are known to stem from difficulty in sensory integration [45, 46], we expected the vestibular group to show more postural sway with magnified differences from the other groups in the most challenging task. Indeed, differences in postural sway between controls and participants with vestibular hypofunction were magnified as the sensory challenge increased. However, we also observed differences between the vestibular group and

controls on the static/ stable scene. These differences are larger than we observed in prior studies [47] possibly because these patients were recruited prior to vestibular rehabilitation.

## What did we learn about postural responses in individuals with USNHL?

While our USNHL sample was small, their postural behavior was consistent, and suggests new hypotheses for future research. The USNHL group did not increase their postural sway in the presence of visual perturbations, particularly when standing on a compliant surface. This behavior could suggest that people with USNHL overly rely on somatosensory input and not on vision, consistent with [23] that found increased somatosensory reliance in deaf individuals compared to controls [48]. observed significant gray matter volume increases in the somatosensory and motor systems as well as decreases in the auditory and visual systems in individuals with single-sided hearing loss further supporting this hypothesis It is also possible that people with USNHL have an overall reduced reaction to any perturbation to avoid loss of balance. This hypothesis is supported by the finding that participants with USNHL did not increase their medio-lateral sway as much as controls with the addition of an unstable surface. Studies of standing balance, typically consider increased postural sway as indication of poor balance. However, some increase in sway with visual and somatosensory perturbations is expected in healthy adults as part of an adaptive automatic control of standing balance [49]. Little to no change in sway with sensory perturbations is thought to be related to overly conscious movement processing or attempting to cognitively control a process otherwise considered automatic. Such behaviors have been observed in adults with phobic postural vertigo [50], healthy adults tested on an elevated platform [51], and children with autistic spectrum disorders [52]. This new hypothesis calls for further investigation of balance in people with USNHL.

## Why didn't sound matter?

Recent studies suggest that healthy individuals use auditory cues for postural control [21, 53–56]. The lack of differences in sound conditions in this study should be interpreted in the context of the specific sound stimulus used and may not speak to the overall impact of auditory cues in postural control. To create the auditory stimulus for this study, we used broadband sounds, known to influence postural sway more than pure tones [9]. We designed amplitude-modulated white noise, with a non-spatialized noise source. This rhythmic sound was designed to mimic the well-established [26] visual perturbation but at a distinct, different frequency such that the sounds and visuals were not fully aligned. We expected that auditory perturbations would increase postural sway in healthy controls on the most challenging task and magnify postural sway in vestibular hypofunction in all conditions. It is possible that the cycling white noise was neither moving enough to provide a disturbance nor static enough to provide an auditory anchor. Future studies on ecologically valid sounds may be valuable to better understand how sounds are used for balance.

## Limitations

The sample size is small and the results, especially the novel result regarding the USNHL group, need to be interpreted with caution and replicated in a larger study before drawing clinical conclusions. In our sample individuals with USNHL have had the condition for years whereas half of the vestibular group had symptoms for over 3 months but less than 12. This difference in chronicity should be controlled for in future studies. All participants in the control group reported no hearing impairment but it was not formally tested. While it is possible that older controls had age-related high frequency hearing loss, this study focused on

significant single sided hearing loss which is highly unlikely to be undiagnosed. Also, no diagnostic vestibular testing was done on participants in the control or USNHL groups. However, neither group reported vestibular complaints, and both had comparable outcomes on standardized self-report assessment of balance (ABC) and vestibular dysfunction (DHI). Although the groups were not significantly different in age, it is still a possibility that age had some influence. We provided estimated differences between groups, but it is currently unknown what a clinically important difference is for postural sway measures.

## Conclusions

This pilot study aimed to identify postural strategies in response to sensory perturbations (visual, auditory, somatosensory) in with individuals with USNHL (and no vestibular complaints), individuals with vestibular hypofunction, and age-matched controls. While rhythmic sounds did not appear to make a difference in this specific study, this may relate to the specifics of the sound paradigm tested; as many other studies suggest the importance of sounds to postural control, future studies will further explore other forms of sounds, such as ecologically valid, spatialized moving sounds. Patients with vestibular hypofunction were destabilized by the visual and surface perturbations more than healthy controls and overall had larger postural sway even on the static scene, evident in both center-of-pressure and head movement. Head movement was similar between USNHL and controls, but notably large in the vestibular group, adding to prior data suggesting a specific pattern of head kinematics that is unique to vestibular dysfunction. Future studies should examine whether these patterns change following vestibular rehabilitation and their implications to functional complaints. The USNHL group did not change with sensory perturbations, particularly visual, as much as controls did. It is possible that individuals with USNHL employed a compensatory strategy of conscious control of balance, but these findings need to be replicated in larger studies.

## Supporting information

**S1 Video. Video of moving visuals, rhythmic sounds scenes.**
(MP4)

**S1 Table. Analysis confirming removal of sound as a predictor.**
(DOCX)

**S2 Table. Model point estimates and their 95% Confidence Intervals [CI] per outcome, group and condition.**
(DOCX)

## Author Contributions

**Conceptualization:** Anat V. Lubetzky, Jennifer L. Kelly, Agnieszka Roginska, Bryan D. Hujsak, Maura Cosetti.

**Data curation:** Daphna Harel.

**Formal analysis:** Daphna Harel.

**Funding acquisition:** Anat V. Lubetzky, Jennifer L. Kelly, Daphna Harel, Maura Cosetti.

**Investigation:** Anat V. Lubetzky, Zhu Wang, Ken Perlin, Maura Cosetti.

**Methodology:** Anat V. Lubetzky, Daphna Harel, Agnieszka Roginska, Zhu Wang, Ken Perlin.

**Project administration:** Anat V. Lubetzky, Bryan D. Hujsak, Ken Perlin.

**Resources:** Bryan D. Hujsak, Zhu Wang, Ken Perlin, Maura Cosetti.

**Software:** Agnieszka Roginska, Zhu Wang, Ken Perlin.

**Supervision:** Anat V. Lubetzky, Bryan D. Hujsak, Ken Perlin.

**Visualization:** Anat V. Lubetzky, Daphna Harel.

**Writing – original draft:** Anat V. Lubetzky.

**Writing – review & editing:** Jennifer L. Kelly, Daphna Harel, Agnieszka Roginska, Bryan D. Hujsak, Zhu Wang, Ken Perlin, Maura Cosetti.

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
