## [Decision Letter · Decision Letter 0]

6 Jul 2022

PONE-D-22-14616Insight into postural control patterns in unilateral sensorineural hearing loss, vestibular hypofunction and age-matched controlsPLOS ONE

Dear Dr. Lubetzky,

Thank you for submitting your manuscript to PLOS ONE. After careful consideration, we feel that it has merit but does not fully meet PLOS ONE’s publication criteria as it currently stands. Therefore, we invite you to submit a revised version of the manuscript that addresses the points raised during the review process.

The reviewers raised some important questions regarding your methods and findings. Please address the comments made by the reviewers through an itemized rebuttal letter.

Please make all your data available (as required by the journal) or provide an explanation on why some of the data cannot be shared.

We look forward to receiving your revised manuscript.

Kind regards,

Rafael da Costa Monsanto, M.D.

Academic Editor

PLOS ONE

Journal Requirements:

8. Please include your tables as part of your main manuscript and remove the individual files. Please note that supplementary tables (should remain/ be uploaded) as separate "supporting information" files.

10. We note that Figure 1 includes an image of a participant in the study.

Reviewers' comments:

Reviewer's Responses to Questions

**Comments to the Author**

1. Is the manuscript technically sound, and do the data support the conclusions?

Reviewer #1: Partly

Reviewer #2: Yes

Reviewer #3: Partly

2. Has the statistical analysis been performed appropriately and rigorously? 

Reviewer #1: Yes

Reviewer #2: Yes

Reviewer #3: I Don't Know

3. Have the authors made all data underlying the findings in their manuscript fully available?

Reviewer #1: Yes

Reviewer #2: Yes

Reviewer #3: Yes

4. Is the manuscript presented in an intelligible fashion and written in standard English?

Reviewer #1: Yes

Reviewer #2: Yes

Reviewer #3: Yes

5. Review Comments to the Author

Reviewer #1: I read the manuscript entitled “Insight into postural control patterns in unilateral sensorineural hearing loss, vestibular hypo function and age-matched controls” with great interest.

I would like to acknowledge the time and effort the authors put into writing such an interesting manuscript.

Major issues:

1. Considering that the authors aimed to compare multi sensory integration between individuals with USNHL ( and no vestibular complaints), individuals with vestibular hypo function and age-matched controls , I suggest adding a few extra lines to describe and compare causes of hearing loss and vestibular dysfunction, classification of hearing loss (sensorineural or mixed/conductive hearing loss, severity of hearing loss) and vestibular dysfunction.

2. All participants in the control group reported no hearing impairment and no vestibular symptoms but it was not formally tested. This information is important and should be added to the manuscript.

3. USNHL group was not submitted to vestibular tests. This information is important and should be added to the manuscript.

Reviewer #2: This “PONE-D-22-14616” manuscript intitle “Insight into postural control patterns in unilateral sensorineural hearing loss, vestibular hypofunction and age-matched control” is a study developed to identify postural strategies in response to sensory perturbations (visual, auditory, somatosensory). The article compares multisensory integration between individuals with unilateral hearing loss (and no vestibular complaints), individuals with vestibular hypofunction and age-matched controls in a well-established protocol using virtual reality glasses where auditory stimuli were added.

I congratulate the initiative of this clinical study which, despite having an extremely restricted number of volunteers and patients in each group, is well written with a clear design that carefully used different auditory and positional sensory tests to verify the postural sway responses in unilateral hearing loss and vestibular hypofunction.

Because it is a pilot study whose limitations are well-punctuated, and the conclusions only suggest possible patterns of compensatory responses that lead to future studies that can confirm this insight; this manuscript has become very extensive with figures and tables that overlap. I suggest checking the need for all figures, tables and video.

Reviewer #3: Although the manuscript is generally interesting, I have a few concerns:

1) Title:

- The title is too wordy. I would recommend describing only the main objective of the study, without including methods (e.g., age-matched controls). .

2) Purpose of the study

- There is a small inconsistency between the purpose of the study between the abstract (..."identify postural strategies in response to sensory perturbations") and the body of the manuscript ("compare multisensory integration ... and characterize postural responses to sensory perturbations").

3) Material and methods:

- It is important to characterize the Hearing Loss using validated methods, such as WHO,2020 (https://www.who.int/publications/i/item/basic-ear-and-hearing-care-resource) or even a different one, and not only based on the asymmetry article that was referenced. Furthermore, the reference used to justify these methods is quite old

- Control participants self-reported normal hearing and absence of vestibular symptoms. As they were not evaluated before the beginning of the study, this could be considered a shortcoming of your study and thereby could have compromised the validity of some of your findings. Can you further clarify?

- In the table it is written that the patients have a unilateral hearing loss for years and this may imply a better use of the other somatosensory cues. Patients with vestibular hypofunction have had these symptoms for a much shorter time as compared with the hearing loss, which could have deemed a more important vestibular symptomatology as compared with the hearing loss.

- As this is a pilot study, these data (and further variables) could have been better controlled.

4) Results

- A statistical analysis could be performed between the three groups or between USNHLxVestibular groups?

- The results of the questionnaires are described in the tables, but I do not see them discussed in the text.

5) Conclusion:

- The conclusions must be reformulated in view of the objectives.

6. PLOS authors have the option to publish the peer review history of their article (what does this mean?). If published, this will include your full peer review and any attached files.

Reviewer #1: **Yes: **Thais Gomes Abrahao Elias

Reviewer #2: No

Reviewer #3: No

---

## [Author Response · Author response to Decision Letter 0]

18 Jul 2022

Summary of Revisions

PONE-D-22-14616

Insight into postural control in unilateral sensorineural hearing loss and vestibular hypofunction 

PLOS ONE

Dear Dr. Monsanto and fellow referees, 

We thank you for a thorough, detailed, constructive and very helpful review. We have made all of the corrections and revised the manuscript according to your suggestions. Below is a point-by-point list detailing how each of your comments was addressed. 

Sincerely,

The Authors

Journal Requirements:

Done.

Done.

Done. Participants provided a written informed consent. No minors were included in the study.

Grant numbers were added as applicable. 

The data are available here:

https://data.mendeley.com/datasets/kcvbs468rm/1

To cite this dataset:

Lubetzky, Anat (2022), “Hearing Health Foundation ERG project”, Mendeley Data, V1, doi: 10.17632/kcvbs468rm.1

The statement was removed and a supplementary file (S1 Table) showing the results of the analysis for 2 levels of sounds was added.

The statement was expanded and clarified as follows:

This study was approved by the Institutional Review Board of the Icahn School of Medicine at Mount Sinai (#STUDY-18-00431) and New York University Committee on Activities Involving Research Subjects (#IRB-FY2016-155). All participants first signed a written informed consent.

8. Please include your tables as part of your main manuscript and remove the individual files. Please note that supplementary tables (should remain/ be uploaded) as separate "supporting information" files.

Done. 

Done.

10. We note that Figure 1 includes an image of a participant in the study.

The 2 individuals in the figure signed the PLOS consent and the statement was added to the Methods. 

 

Reviewers' comments:

Reviewer #1: I read the manuscript entitled “Insight into postural control patterns in unilateral sensorineural hearing loss, vestibular hypo function and age-matched controls” with great interest.

I would like to acknowledge the time and effort the authors put into writing such an interesting manuscript.

Thank you!! �

Major issues:

1. Considering that the authors aimed to compare multi-sensory integration between individuals with USNHL (and no vestibular complaints), individuals with vestibular hypo function and age-matched controls, I suggest adding a few extra lines to describe and compare causes of hearing loss and vestibular dysfunction, classification of hearing loss (sensorineural or mixed/conductive hearing loss, severity of hearing loss) and vestibular dysfunction.

The following statements were added / expanded:

• Included patients had either with unilateral peripheral vestibular hypofunction and no hearing loss, i.e., vestibular neuritis, or sudden idiopathic unilateral sensorineural hearing loss (USNHL). 

• Participants in the USNHL were excluded for mixed or conductive hearing loss. 

Details regarding the severity of HL appear in the Table 3 and description of the sample (minimum moderate unilateral HL of 41 dB 4 frequency PTA or worse).

2. All participants in the control group reported no hearing impairment and no vestibular symptoms but it was not formally tested. This information is important and should be added to the manuscript.

This statement was added to the results section under ‘sample’.

3. USNHL group was not submitted to vestibular tests. This information is important and should be added to the manuscript.

This statement was added to the results section under ‘sample’.

Reviewer #2: This “PONE-D-22-14616” manuscript intitle “Insight into postural control patterns in unilateral sensorineural hearing loss, vestibular hypofunction and age-matched control” is a study developed to identify postural strategies in response to sensory perturbations (visual, auditory, somatosensory). The article compares multisensory integration between individuals with unilateral hearing loss (and no vestibular complaints), individuals with vestibular hypofunction and age-matched controls in a well-established protocol using virtual reality glasses where auditory stimuli were added.

I congratulate the initiative of this clinical study which, despite having an extremely restricted number of volunteers and patients in each group, is well written with a clear design that carefully used different auditory and positional sensory tests to verify the postural sway responses in unilateral hearing loss and vestibular hypofunction.

Thank you! �

Because it is a pilot study whose limitations are well-punctuated, and the conclusions only suggest possible patterns of compensatory responses that lead to future studies that can confirm this insight; this manuscript has become very extensive with figures and tables that overlap. I suggest checking the need for all figures, tables and video.

To reduce overlap Table 4 was moved to a supplementary file (S2 Table).

Reviewer #3: Although the manuscript is generally interesting, I have a few concerns:

1) Title:

- The title is too wordy. I would recommend describing only the main objective of the study, without including methods (e.g., age-matched controls). .

The title was shortened as per your suggestion.

2) Purpose of the study

- There is a small inconsistency between the purpose of the study between the abstract (..."identify postural strategies in response to sensory perturbations") and the body of the manuscript ("compare multisensory integration ... and characterize postural responses to sensory perturbations").

We changed the purpose in the intro to make sure the wording is consistent. 

3) Material and methods:

- It is important to characterize the Hearing Loss using validated methods, such as WHO,2020 (https://www.who.int/publications/i/item/basic-ear-and-hearing-care-resource) or even a different one, and not only based on the asymmetry article that was referenced. Furthermore, the reference used to justify these methods is quite old

The definition was updated to reflect most recent WHO guidelines with the relevant reference. 

- Control participants self-reported normal hearing and absence of vestibular symptoms. As they were not evaluated before the beginning of the study, this could be considered a shortcoming of your study and thereby could have compromised the validity of some of your findings. Can you further clarify?

The following clarification was added to the limitations section:

While it is possible that older controls had age-related high frequency hearing loss, this study focused on significant single sided hearing loss which is highly unlikely to be undiagnosed….

…. neither group reported vestibular complaints, and both had comparable outcomes on standardized self-report assessment of balance (ABC) and vestibular dysfunction (DHI).

- In the table it is written that the patients have a unilateral hearing loss for years and this may imply a better use of the other somatosensory cues. Patients with vestibular hypofunction have had these symptoms for a much shorter time as compared with the hearing loss, which could have deemed a more important vestibular symptomatology as compared with the hearing loss.

Indeed, in our sample individuals with USNHL have had the condition for years whereas half of the vestibular group had symptoms for over 3 months but less than 12. This could potentially influence their ability to cope with sensory perturbations and as per your comment we now added this point to the limitations section. In vestibular research, chronic vestibular dysfunction is considered 3 months or over because spontaneous recovery is not expected beyond 3 months. Patients with single sided hearing loss however, either experience it since childhood or undergo a series of injections prior to concluding that their loss cannot be resolved. 

Descriptively, when comparing within the vestibular group those that had symptoms less than 1 year (N=6) vs. longer than 1 year (N=6) we observed that these groups were comparable on age, DHI, ABC and all sway parameters except for DP AP on the most challenging condition (foam, dynamic vision) where the vestibular group of over 1 year had significantly less sway than the vestibular group of 1 year or under. While this observation suggests that differences in chronicity could have influenced sensory integration strategies and so should be considered in future research note that the 1 year or over group was still significantly higher than the HL group. 

- As this is a pilot study, these data (and further variables) could have been better controlled.

4) Results

- A statistical analysis could be performed between the three groups or between USNHLxVestibular groups?

The models that we ran performed an analysis of the 3 groups using contrast coding where the control group was used as the reference. In our results, we found that the vestibular group consistently showed larger sway (COP and head) than the controls whereas the USNHL either did not differ from controls or in fact showed less change in sway with sensory perturbations. Therefore, we concluded that a direct comparison of the vestibular group to the USNHL group is not needed. 

- The results of the questionnaires are described in the tables, but I do not see them discussed in the text.

Details were added under Results – Sample.

5) Conclusion:

- The conclusions must be reformulated in view of the objectives.

We shortened and reorganized the conclusion to focus directly on the purpose of this study.

---

## [Decision Letter · Decision Letter 1]

4 Oct 2022

Insight into postural control in unilateral sensorineural hearing loss and vestibular hypofunction

PONE-D-22-14616R1

Dear Dr. Lubetzky,

We’re pleased to inform you that your manuscript has been judged scientifically suitable for publication and will be formally accepted for publication once it meets all outstanding technical requirements.

Kind regards,

Rafael da Costa Monsanto, M.D.

Academic Editor

PLOS ONE

Additional Editor Comments (optional):

Reviewers' comments:

Reviewer's Responses to Questions

**Comments to the Author**

1. If the authors have adequately addressed your comments raised in a previous round of review and you feel that this manuscript is now acceptable for publication, you may indicate that here to bypass the “Comments to the Author” section, enter your conflict of interest statement in the “Confidential to Editor” section, and submit your "Accept" recommendation.

Reviewer #2: All comments have been addressed

Reviewer #3: All comments have been addressed

2. Is the manuscript technically sound, and do the data support the conclusions?

Reviewer #2: Yes

Reviewer #3: Yes

3. Has the statistical analysis been performed appropriately and rigorously? 

Reviewer #2: Yes

Reviewer #3: Yes

4. Have the authors made all data underlying the findings in their manuscript fully available?

Reviewer #2: Yes

Reviewer #3: Yes

5. Is the manuscript presented in an intelligible fashion and written in standard English?

Reviewer #2: Yes

Reviewer #3: Yes

6. Review Comments to the Author

Reviewer #2: Thank you for thoroughly addressing the comments. The authors have provided corresponding information and the manuscript has improved overall.

Reviewer #3: (No Response)

7. PLOS authors have the option to publish the peer review history of their article (what does this mean?). If published, this will include your full peer review and any attached files.

Reviewer #2: No

Reviewer #3: No

---

## [Editor Report · Acceptance letter]

6 Oct 2022

PONE-D-22-14616R1 

Insight into postural control in unilateral sensorineural hearing loss and vestibular hypofunction 

Dear Dr. Lubetzky:

I'm pleased to inform you that your manuscript has been deemed suitable for publication in PLOS ONE. Congratulations! Your manuscript is now with our production department. 

Kind regards, 

on behalf of

Dr. Rafael da Costa Monsanto 

Academic Editor

PLOS ONE